# Long-Term Outcomes of Endoscopic Submucosal Dissection for Superficial Esophageal Squamous Cell Carcinoma

**DOI:** 10.3390/cancers12102849

**Published:** 2020-10-02

**Authors:** Toshihiro Nishizawa, Hidekazu Suzuki

**Affiliations:** 1Department of Gastroenterology and Hepatology, International University of Health and Welfare, Narita Hospital, Narita 286-8520, Japan; nisizawa@kf7.so-net.ne.jp; 2Division of Gastroenterology and Hepatology, Department of Internal Medicine, Tokai University School of Medicine, Isehara 259-1193, Japan

**Keywords:** ESD, long-term outcomes, superficial esophageal cancer

## Abstract

**Simple Summary:**

Endoscopic submucosal dissection (ESD) was developed to enhance en bloc resections and was applied to esophageal lesions. ESD showed advantages over endoscopic mucosal resection (EMR) regarding the en bloc resection rate, complete resection rate, and local recurrence rate. In this review, we summarize up-to-date reports with 5-year survival after ESD for superficial esophageal squamous cell carcinoma. ESD has shown excellent long-term outcomes for superficial esophageal squamous cell carcinoma with epithelium or lamina propria invasion. ESD would be the first choice for superficial esophageal squamous cell carcinoma without obvious submucosal invasion, although more than three-fourths circumferential resection could induce post-operative stenosis.

**Abstract:**

In this review, we summarize up-to-date reports with 5-year survival after endoscopic submucosal dissection (ESD) for superficial esophageal squamous cell carcinoma. In ESD for the depth of the epithelium (m1) or lamina propria (m2), the 5-year cause-specific survival and 5-year overall survival rates were reported to be 98–100%, and 85–95%, respectively. In cases with submucosal invasion or vascular involvement, additional prophylactic treatment such as chemoradiotherapy or surgery was recommended, and the 5-year cause-specific survival and 5-year overall survival rates were reported to be 85–100%, and 56–84%, respectively. Additional treatment might be too invasive for the elderly or patients with severe comorbidities. The risk of additional therapy should be balanced against the risk of lymph node metastasis, considering the life expectancy of such patients.

## 1. Introduction

Endoscopic resection is becoming the main procedure for mucosal esophageal cancer [1,2]. In 1990, Inoue et al. developed the endoscopic cap resection technique for esophageal lesions [3]. However, widespread lesions are hard to be resected en bloc. Endoscopic submucosal dissection (ESD) was developed to enhance en bloc resections [4,5] and has been applied to esophageal lesions in Japan [6] (Figure 1). ESD showed advantages over endoscopic mucosal resection (EMR) regarding the en bloc resection rate, complete resection rate, and local recurrence rate [7,8]. In 2008, esophageal ESD was approved by the Ministry of Health, Labor, and Welfare of Japan and obtained a code for reimbursement. The technique and devices for ESD have improved [9,10,11,12,13,14,15] and seem to have matured [16,17,18,19]. Considerable studies evaluating long-term outcomes after ESD for superficial esophageal cancer have been gathered [20,21,22,23]. In this review, we summarize up-to-date reports with 5-year survival after ESD for superficial esophageal squamous cell carcinoma.

## 2. Literature Search and Selection

The literature was systematically searched using PubMed, Cochrane Library, and the Igaku–Chuo–Zasshi database in Japan (up to August 2020). The search words were: (esophageal) AND (endoscopic submucosal dissection) AND (squamous cell carcinoma) AND (long-term outcomes). The inclusion criteria were: (1) study design: cohort study, (2) participants: patients who had esophageal squamous cell carcinoma, (3) intervention: ESD, and (4) outcome: 5-year survival. The exclusion criteria were: (1) studies limited to large lesions, (2) 5-year survival including EMR, (3) rescue ESD after chemoradiotherapy, (4) meeting abstracts, (5) duplication, and (6) review articles. One hundred and fifty-two citations were retrieved by the literature search process. Among these, we excluded 116 studies according to the exclusion criteria (36 unrelated topics, 63 meeting abstracts, 1 duplication, and 16 review articles). The remaining 36 studies were scrutinized and 23 studies were rejected (14 studies without 5-year survival, 2 studies limited to large lesions, 5 studies including EMR, and 2 rescue ESD after chemoradiotherapy). Due to the limited number of studies comparing ESD and chemoradiotherapy, we allowed the inclusion of EMR for that study. We included 13 studies in our final review. The flowchart showing the search process is shown in Figure 2.

## 3. Long-Term Outcomes of ESD for Superficial Esophageal Squamous Cell Carcinoma

The frequency of lymph node metastasis varies widely depending on the depth of invasion. The reported rates of lymph node metastasis are 0.0%–5.6% for the depth of the epithelium (m1) or lamina propria (m2), 8.0%–18.0% for the depth of the muscularis mucosae (m3), 11.0%–53.1% for submucosal invasion to 200 μm or less (sm1), and 30.0%–53.9% for deep submucosal invasion exceeding 200 μm (sm2 or sm3) [24,25,26]. If the pathology after esophageal ESD shows submucosal invasion, additional prophylactic treatment such as chemoradiotherapy or surgery is recommended. The Japan Esophageal Society guidelines recommend additional prophylactic treatment for sm1 or more or vascular invasion [27]. The European Society of Gastrointestinal Endoscopy (ESGE) guidelines recommend additional treatment for sm2 or more, vascular involvement, poorly differentiated tumors, or positive vertical margins [28]. Several studies have evaluated long-term outcomes with depth stratification (Table 1).

Ono et al. performed esophageal ESD in 84 patients from 2002 to 2008 and the rates of en bloc resection and complete resection were 100% and 88%, respectively [20]. The perforation rate was 4%. They also evaluated long-term outcomes. The median observation period was 632 days (range: 8–2358 days). High-grade intraepithelial neoplasm (HGN) was also included. The depth stratification was HGN/m1/m2 and m3/sm1/sm2. Additional prophylactic treatment was recommended if m3–submucosal (sm) invasion or vascular involvement was detected in the specimens after ESD. In the m3/sm1/sm2 group, six patients (21%) received chemoradiotherapy or radiotherapy and nine patients (32%) underwent surgery (operation) as additional therapy. During the observation period, local recurrence occurred in one patient with m1 and was successfully treated with rescue ESD. Distant metastases occurred in two patients with m3. Three patients died of esophageal cancer and seven patients died of other causes. The 5-year overall survival rates of the HGN/m1/m2 group and m3/sm1/sm2 group were 95% and 56%, respectively. The 5-year cause-specific survival rates of the HGN/m1/m2 group and m3/sm1/sm2 group were 100% and 85%, respectively.

Tanaka et al. performed esophageal ESD in 84 patients from 2005 to 2011 and the rates of en bloc resection and complete resection were 98.8% and 83.1%, respectively [21]. The perforation rate was 1.6%. The depth stratification was m1/m2, m3/sm1, and sm2. Additional prophylactic treatment was recommended if sm invasion, vascular involvement, or a positive vertical margin was detected in the specimens after ESD. In the m3/sm1 group, twelve patients (21%) received chemoradiotherapy or radiotherapy and one patient (2%) underwent surgery as additional therapy. In the sm2 group, fourteen patients (54%) received chemoradiotherapy or radiotherapy and three patients (12%) underwent surgery. During the observation period, local recurrence occurred in two patients with m2, and lymph node metastasis occurred in six patients (10.7%) with m3/sm1 and in two patients (7.7%) with sm2. Three patients died of esophageal cancer and sixteen patients died of other causes. The 5-year overall survival rates of the m1/m2, m3/sm1 and sm2 groups were 85%, 82%, and 83%, respectively. The 5-year cause-specific survival rates of the m1/m2, m3/sm1, and sm2 groups were 98%, 93%, and 100%, respectively. In the m1/m2 group, death due to esophageal cancer occurred in one patient who had metachronous esophageal cancer with sm2. The patient received chemoradiotherapy but died due to the metastasis.

Nagami et al. performed esophageal ESD in 84 patients from 2006 to 2009 and the rates of en bloc resection and complete resection were 100% and 90.4%, respectively [22]. The perforation rate was 0%. The median observation period was 72.9 months (interquartile range: 63.0–85.7 months). HGN was also included. The depth stratification was HGN/m1/m2, m3/sm1, and sm2. Additional prophylactic treatment was recommended if m3–sm invasion, vascular involvement, or a positive vertical margin was detected in the specimens after ESD. In the m3/sm1 group, thirteen patients (68%) received chemoradiotherapy or chemotherapy, and two patients (11%) underwent surgery as additional therapy. In the sm2 group, three patients (75%) received chemoradiotherapy or chemotherapy, and one patient (25%) underwent surgery. During the observation period, there was no local recurrence or distant metastasis. The causes of death included pneumonia (*n* = 3), lung cancer (*n* = 2), gastric cancer (*n* = 2), laryngeal cancer (*n* = 1), pancreatic cancer (*n* = 1), adverse effects of radiotherapy (*n* = 3), and other causes (*n* = 2). The 5-year overall survival rates of the m1/m2, m3/sm1, and sm2 groups were 95%, 84%, and 75%, respectively. The 5-year cause-specific survival rate was 100%.

Takahashi et al. investigated the long-term outcome of ESD for esophageal squamous cell carcinoma with m3/sm1 [23]. Additional prophylactic treatment was recommended if pathological vascular involvement was detected in the specimens after ESD. Endoscopy and computed tomography (CT) were performed as much as possible every six months after ESD. One hundred and two patients underwent esophageal ESD; vascular involvement was found in twenty-one patients (20.6%). Among these, nine patients (8.8%) received chemoradiotherapy and three patients (2.9%) underwent surgery. The median observation period was 71.5 months (range: 9–144 months). During the observation period, local recurrence occurred in one patient. Lymph node metastasis or distant metastasis occurred in 12 patients (11.8%), 11 of whom underwent salvage therapy. Two patients died of esophageal cancer and twenty-four patients died of other causes. The 5-year overall survival rates and cause-specific survival rates were 84.1% and 97.5%, respectively.

To summarize, the strategy of ESD and additional treatment depending on the depth of invasion showed an excellent 5-year cause-specific survival rate. The 5-year overall survival rates were relatively low, especially for m3 or submucosal invasion. This discrepancy may be due to the invasiveness of the additional treatment.

## 4. Long-Term Outcomes of ESD for Superficial Esophageal Squamous Cell Carcinoma without Additional Treatment

Additional treatment such as chemoradiotherapy or surgery could be too invasive for the elderly or patients with severe comorbidities. There are two studies evaluating the long-term outcomes of ESD for superficial esophageal squamous cell carcinoma without additional treatment (Table 2).

Joo et al. performed esophageal ESD in 28 patients from 2005 to 2012. Pre-operative endoscopic ultrasound (EUS) and CT were routinely performed. The rates of en bloc resection and complete resection were 93% and 86%, respectively [29]. The perforation rate was 7%. The mean age was 64 years (range: 46–76 years). The mean observation period was 23 months (range: 2–78 months). Five patients (18%) had m3/sm1/sm2 and were recommended for additional treatment. However, all of them refused additional treatment. Pathological vascular involvement was absent in all cases. Local recurrence occurred in two patients with m3–sm2 and was treated with surgery or rescue ESD. None died of esophageal cancer and the 5-year overall survival rate was 84%.

Qi et al. reported 158 cases of ESD without additional therapy. Pre-operative EUS and CT were routinely performed. The study excluded 10 cases of ESD with additional treatment [30]. The rate of complete resection was 99.4%. The mean age was 66 years (range: 60–84 years). The median observation period was 56 months (interquartile range: 47–68 months). Sixty-nine patients (44%) had m3–sm1. Among these, six patients (8.7%) had pathological vascular involvement and one patient died of esophageal cancer due to metastatic recurrence. The 5-year overall survival rates of the m1/m2 group and m3/sm1 group were 96.6% and 95.6%, respectively.

In the study by Joo et al., the pathological vascular involvement rate was 0%. The study by Qi et al. excluded cases with additional treatment. These points could bias the results, and the very good long-term outcomes might be overestimated. However, a strategy that does not force additional treatment would be promising, especially for the elderly or patients with severe comorbidities.

## 5. Comparison between Elderly and the Younger Generation in Esophageal ESD

One study compared the long-term outcomes of esophageal ESD between elderly and young generation (Table 3).

Iizuka et al. performed esophageal ESD in 664 consecutive patients between 2008 and 2016 [31]. HGN was also included. Clinical outcomes were compared between those aged 75 years or older (*n* = 162) and those aged younger than 75 years (*n* = 502). Additional prophylactic treatment was recommended if m3–sm invasion was detected in the specimens after ESD. The m3–sm invasion rates for the elderly and the younger group were 21% and 22.1%, respectively. In the elderly group, thirteen patients (8%) received chemoradiotherapy and one patient (0.6%) underwent surgery as additional therapy. In the younger group, seventy-two patients (14.3%) received chemoradiotherapy and thirty-two patients (6.4%) underwent surgery. The rate of additional treatments in the elderly group was significantly lower than in the younger group (*p* < 0.001). The 5-year survival rates and the cause-specific survival rates for the elderly and the younger groups were 83.6% versus 91.2% and 97.3% versus 97.5%. The long-term outcomes of the elderly and the younger groups were comparable, although the rate of additional treatments was low in the elderly.

## 6. Comparison between ESD and Other Resection Methods for Superficial Esophageal Squamous Cell Carcinoma

Before ESD was introduced, the other resection methods were either EMR or surgery [32]. There are several studies comparing the long-term outcomes of ESD with other resection methods (Table 4).

Min et al. compared ESD and surgical resection [33]. Pre-operative EUS and CT were routinely performed. Patients with superficial esophageal squamous cell carcinoma without obvious submucosal invasion were included. Sixteen cases with rescue esophagectomy after ESD were excluded. Propensity score matching was performed to minimize the selection bias. The m3–sm invasion rates for the ESD group and the surgery group were 46.7% and 47.5%, respectively. The 5-year overall survival rates and disease-free survival rates for the ESD group and the surgery group were 93.9% versus 91.2% and 92.8% versus 95.3%. The early adverse event rates for the ESD group and the surgery group were 8.9% versus 48.2% (*p* < 0.001). The early adverse events in the ESD group were micro-perforation or frank perforation. Early adverse events in the surgery group included a pulmonary event, a cardiovascular event, vocal cord palsy, wound infection, anastomotic leakage, and so on.

Lee et al. also compared ESD and surgical resection [34]. Pre-operative EUS and CT were routinely performed. Patients with superficial esophageal squamous cell carcinoma without lymph node metastasis were included. Five cases with rescue esophagectomy after ESD were excluded. Propensity score matching was performed to minimize the selection bias. The m3–sm invasion rates for the ESD group and the surgery group were 44.1% and 41.2%, respectively. The 5-year overall survival rates and disease-free survival rates for the ESD group and the surgery group were 89.4% versus 87.8% and 90.9% versus 91.6%. The early major adverse event rates for the ESD group and the surgery group were 2.9% versus 23.5% (*p* < 0.001). The ESD group had shorter hospital stays (median: 3.0 days vs. 16.5 days; *p* < 0.001) than the surgery group. These studies concluded that the long-term outcomes of ESD and surgery were comparable and ESD was less invasive.

Berger et al. compared ESD and EMR [35]. Pre-operative EUS was routinely performed. The m3–sm invasion rates for the ESD group and the EMR group were 32.8% and 33.8%, respectively. The complete resection rates for the ESD group and the EMR group were 97.1% versus 85% (*p* < 0.01). The 5-year disease-free survival rates for the ESD group and the EMR group were 95.2% versus 73.4% (*p* < 0.01). The perforation rates for the ESD group and the EMR group were 2.9% versus 1.3%, respectively. The long-term outcomes of ESD were better than those of EMR for superficial esophageal squamous cell carcinoma. Thus ESD would be the first choice for superficial esophageal squamous cell carcinoma without obvious submucosal invasion.

## 7. Chemoradiotherapy and ESD for Esophageal Squamous Cell Carcinoma with Submucosal Invasion

For esophageal squamous cell carcinoma with submucosal invasion, surgery is the standard treatment if the patients have surgical tolerability. If the patients do not have surgical tolerability, the less invasive alternatives are chemoradiotherapy or radiotherapy [36,37]. Recently, endoscopic resection combined with chemoradiotherapy has been suggested as a new alternative [38,39].

Yoshimizu et al. compared definitive chemoradiotherapy and endoscopic resection (ESD/EMR) combined with chemoradiotherapy in patients with esophageal squamous cell carcinoma with submucosal invasion [38] (Table 5). Pre-operative EUS and CT were routinely performed. The radiation dose of definitive chemoradiotherapy was 60 Gy; that of endoscopic resection combined with chemoradiotherapy was reduced to 41.4–50.4 Gy. The 5-year overall survival rates for the definitive chemoradiotherapy group and the endoscopic resection–chemoradiotherapy group were 79.1% and 85.1%, respectively. The 5-year disease-free survival rates for the definitive chemoradiotherapy group and the endoscopic resection–chemoradiotherapy group were 59.2% and 85.1%, respectively. The 5-year disease-free survival rate of the endoscopic resection–chemoradiotherapy group was significantly higher than that of the definitive chemoradiotherapy group (*p* < 0.05). The local recurrence rates were 19% and 0% and the metastatic recurrence rates were 7% and 10%, for the definitive chemoradiotherapy group and the endoscopic resection–chemoradiotherapy group, respectively. The local recurrence rate of the endoscopic resection–chemoradiotherapy group was significantly lower than that of the definitive chemoradiotherapy group (*p* < 0.05). Endoscopic resection combined with chemoradiotherapy showed better outcomes than definitive chemoradiotherapy. ESD combined chemoradiotherapy would be a promising option for esophageal squamous cell carcinoma with a submucosal invasion.

## 8. Cumulative Metachronous Cancer Rate after Esophageal ESD

Metachronous recurrence occasionally occurs after the ESD of esophageal squamous cell carcinoma. There are two studies reporting the 5-year cumulative metachronous cancer rates after esophageal ESD (Table 6). Tsuji et al. reported that the 3- and 5-year cumulative metachronous cancer rates were 9.9% and 24.5%, respectively [40]. Kuwai et al. reported that the 3- and 5-year cumulative metachronous cancer rates were 14% and 26%, respectively [41]. The metachronous cancer rates are considerably high and surveillance endoscopy should be performed carefully. Abstinence from alcohol is also important to prevent recurrence [42,43].

## 9. Conclusions

ESD would be the first choice for superficial esophageal squamous cell carcinoma without obvious submucosal invasion, although more than three-fourths circumferential resection could induce post-operative stenosis [44,45]. ESD has shown excellent long-term outcomes for superficial esophageal squamous cell carcinoma with m1/m2. Additional treatment such as surgery and chemoradiotherapy should be recommended in the case of vascular involvement or submucosal invasion. The selection between surgery and chemoradiotherapy should be made after assessing the patient’s surgical tolerability [36]. Additional treatment might be too invasive for the elderly or patients with severe comorbidities. The risk of additional therapy should be balanced against the risk of lymph node metastasis, considering the life expectancy of such patients.

## Figures and Tables

**Figure 1 cancers-12-02849-f001:**
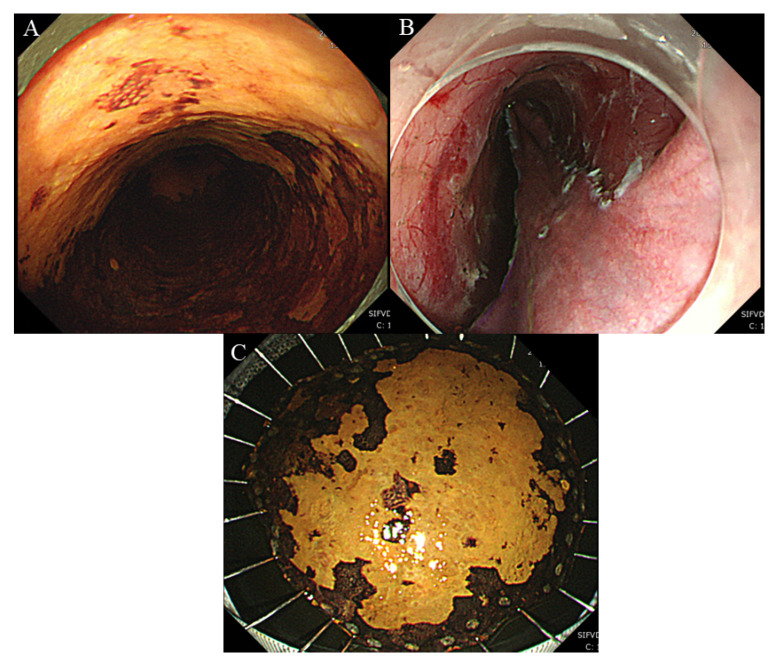
Esophageal endoscopic submucosal dissection (ESD) for superficial esophageal squamous cell carcinoma. (**A**) Widespread superficial esophageal squamous cell carcinoma. (**B**) Post-operative ulcer after ESD. (**C**) Specimen after ESD (size, 50 mm; complete resection).

**Figure 2 cancers-12-02849-f002:**
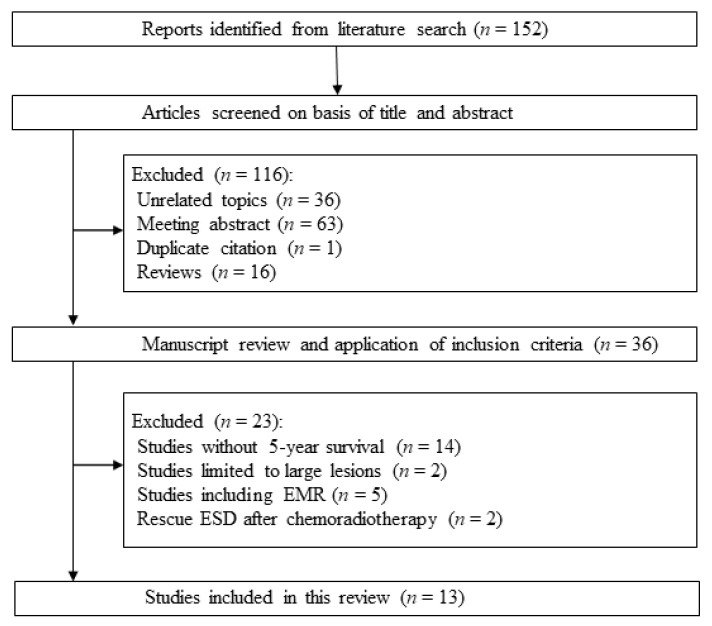
Flow diagram of the systematic literature search.

**Table 1 cancers-12-02849-t001:** The depth of superficial esophageal squamous cell carcinoma and the long-term outcomes of endoscopic submucosal dissection.

Author	Depth	Number of	Additional Therapy	5-Year Overall	5-Year Cause
Year		Patients	Number (%)	Survival	Specific Survival
Ono	HGN/M1/M2	56	-	95%	100%
2009	M3/SM1/SM2	28	CRT/RT 6 (21%), Ope 9 (32%)	56% *	85% *
Tanaka	M1/M2	122	-	85%	98%
2013	M3/SM1	56	CRT/RT 12 (21%), Ope 1 (2%)	82%	93%
	SM2	26	CRT/RT 14 (54%), Ope 3 (12%)	83%	100%
Nagami	HGN/M1/M2	60	-	95%	100%
2017	M3/SM1	19	CRT/CY 13 (68%), Ope 2 (11%)	84%	100%
	SM2	4	CRT/CY 3 (75%), Ope 1 (25%)	75% *	100%
Takahashi	M3/SM1	102	CRT 9 (9%), Ope 3 (3%)	84%	98%
2018					

HGN: High-grade neoplasia; CRT: Chemoradiotherapy; RT: Radiotherapy; CY: Chemotherapy; Ope: Operation (surgery); M1: invasion to the depth of the epithelium; M2: invasion to the depth of the lamina propria: M3: invasion to the depth of the muscularis mucosae; SM1: submucosal invasion to 200 μm or less; SM2: deep submucosal invasion exceeding 200 μm. * *p* < 0.05.

**Table 2 cancers-12-02849-t002:** Long-term outcomes of endoscopic submucosal dissection for superficial esophageal squamous cell carcinoma without additional treatment.

Author	Number of	Age (Years)	Depth	Number of	Vascular	Death Due to	5-Year Overall
Year	Patients	Mean (Range)		Patients (%)	Involvement (%)	Esophageal Cancer	Survival
Joo	28	64 (46–76)	M1/M2	23 (82%)	0	0	84%
2014			M3–SM2	5 (18%)			
Qi	158	66 (60–84)	M1/M2	89 (56%)	3 (3.4%)	0	96.6%
2018			M3–SM1	69 (44%)	6 (8.7%)	1 (1.4%)	95.6%

**Table 3 cancers-12-02849-t003:** Comparison between the elderly and the younger generation after esophageal ESD.

Author	Generation	Number of	m3–sm	Additional Therapy	5-Year Overall	5-Year Cause
Year		Patients	(%)	Number (%)	Survival	Specific Survival
Iizuka	Elderly	162	21%	CRT 13 (8%), Ope 1 (0.6%) ***	83.6% *	97.3%
2019	Young	502	22.1%	CRT 72 (14.3%), Ope 32 (6.4%)	91.2%	97.5%

CRT: Chemoradiotherapy, Ope: Operation (surgery). * *p* < 0.05, *** *p* < 0.001.

**Table 4 cancers-12-02849-t004:** Comparison between endoscopic submucosal dissection and other resection methods for superficial esophageal squamous cell carcinoma.

Author	Therapy	Number of	m3–sm	5-Year Overall	5-Year Disease-Free Survival	Adverse Events
Year		Patients	Number (%)	Survival		
Min	ESD	120	56 (46.7%)	93.9%	92.8%	8.9% ***
2018	Surgery	120	57 (47.5%)	91.2%	95.3%	48.2%
Lee	ESD	34	15 (44.1%)	89.4%	90.9%	2.9% ***
2020	Surgery	34	14 (41.2%)	87.8%	91.6%	23.5%
Berger	ESD	68	26 (32.8%)	-	95.2% **	2.9% †
2019	EMR	80	27 (33.8%)	-	73.4%	1.3% †

EMR: endoscopic mucosal resection, † Perforation only, ** *p* < 0.01, *** *p* < 0.001.

**Table 5 cancers-12-02849-t005:** Endoscopic resection and chemoradiotherapy (CRT) for esophageal squamous cell carcinoma with submucosal invasion.

Author	Treatment	Number of	5-Year Overall	5-Year Disease	Local	Metastatic
Year	(Additional Rate)	Patients	Survival	Free Survival	Recurrence	Recurrence
Yoshimizu	CRT	43	79.1%	59.2% *	19% *	7%
2018	ESD/EMR + CRT	21	85.1%	85.1%	0%	10%

* *p* < 0.05.

**Table 6 cancers-12-02849-t006:** Cumulative metachronous cancer rate after complete resection in esophageal endoscopic submucosal dissection.

Author	Year	Number of	3-Year Metachronous	5-Year Metachronous	Overall
		Patients	Cancer Rate	Cancer Rate	Survival
Tsuji	2015	214	9.9%	24.5%	-
Kuwai	2018	57	14%	26%	75.0%

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
