# Peer review of "Long-Term Outcomes of Endoscopic Submucosal Dissection for Superficial Esophageal Squamous Cell Carcinoma"

_cancers, 2020, doi:10.3390/cancers12102849_

Round 1
Reviewer 1 Report
This is a thoughtful review of studies of ESR with excellent results for ESR for m1 and m2 depths of invasion but for m3 invasion, the disease specific survival was 85-100% but overall 5 year survival was 56=84%. This was attributed to the toxicity of post ESR therapy such as surgery or chemoradiotherapy. This was felt to be due to age and/or comorbidity and yet there was NO data for such a conclusion. The recommendation to not give additional therapy to the elderly or those with comorbidities is a major conclusion but it was be much stronger if data on age/comorbidities was provided
There is no description of how the studies reviewed were selected and whether there were others that were not used. This data should be provided if possible
There was no data on pre-operative staging with endoscopic US or PET which might have detected disease not suitable for ESR. As patients with m3 disease have an over 10% chance of having nodal metastases, perhaps thorough staging would have ruled out those patients for ESR and many might have benefitted from neoadjuvant therapy
Author Response
This is a thoughtful review of studies of ESR with excellent results for ESR for m1 and m2 depths of invasion but for m3 invasion, the disease specific survival was 85-100% but overall 5 year survival was 56=84%. This was attributed to the toxicity of post ESR therapy such as surgery or chemoradiotherapy. This was felt to be due to age and/or comorbidity and yet there was NO data for such a conclusion. The recommendation to not give additional therapy to the elderly or those with comorbidities is a major conclusion but it was be much stronger if data on age/comorbidities was provided
There is no description of how the studies reviewed were selected and whether there were others that were not used. This data should be provided if possible
There was no data on pre-operative staging with endoscopic US or PET which might have detected disease not suitable for ESR. As patients with m3 disease have an over 10% chance of having nodal metastases, perhaps thorough staging would have ruled out those patients for ESR and many might have benefitted from neoadjuvant therapy
The literatures were systematically searched using the PubMed, Cochrane library, and the Igaku-Chuo-Zasshi database in Japan (up to August 2020). The search words were listed below: (esophageal) AND (endoscopic submucosal dissection) AND (squamous cell carcinoma) AND (long-term outcomes). There was no limitation in language. The inclusion criteria were listed below: (1) study design: cohort study; (2) participants: patients who had esophageal squamous cell carcinoma; (3) intervention: endoscopic submucosal dissection (ESD); (4) outcome: 5-year survival. The exclusion criteria were listed below: (1) studies limited to large lesions; (2) 5-year survival including endoscopic mucosal resection (EMR); (3) rescue ESD after chemoradiotherapy (4) meeting abstracts; (5) duplication; and (6) review articles.
As the search results, one hundred fifty-two citations were involved by the literature search process. Among them, we excluded 116 studies according to the exclusion criteria (36 unrelated topics, 63 meeting abstracts, 1 duplication, and 16 review articles). The remaining 36 studies were scrutinized, and 23 studies were rejected (14 studies without 5-year survival, 2 studies limited to large lesions, 5 studies including EMR, 2 rescue ESD after chemoradiotherapy). Due to the limited number of studies comparing ESD and chemoradiotherapy, it was allowed to include EMR for that study. Finally, we included 13 studies in our review. The flowchart showing the searching process is shown in Figure 2. In the revised manuscript, two articles were added (Iizuka et al. in Table 3, and Lee e al. in Table 4).
Pre-operative examinations were different among the included 13 studies. Both pre-operative endoscopic ultrasound (EUS) and computed tomography (CT) were routinely performed in 5 studies. Pre-operative EUS was routinely performed in 1 study. CT were routinely performed in 1 study. In 2 studies, pre-operative EUS was optional. In 2 studies, pre-operative examination included iodine staining or narrow band imaging (NBI). In 2 studies, pre-operative examination was not described. The information about pre-operative EUS and/or CT was added into the revised manuscript. Thank you for your important comments, which were extremely helpful for improving the quality of our manuscript.
Reviewer 2 Report
The authors are summarizing up-to-date reports with 5-year survival after endoscopic submucosal dissection (ESD) for superficial esophageal squamous cell carcinoma. Based on the reviewed data the authors conclude, that "The 5-year overall survival rates in m3 or submucosal invasion were relatively low. This discrepancy may be due to the invasiveness of the additional treatment. Additional treatment could be too invasive for the elderly or patients with severe comorbidities. Considering the balance between the risk of metastasis and life expectancy for the elderly or patients with severe comorbidities, the abandonment of additional treatment might be a suitable choice."
This conclusion is not fully supported by the data presented. Details on the criteria for additional treatment of the studies reviewed are partially not given or not discussed. Which criteria were used within the studies for additional treatment? Patient specific individual treatment strategies? standardized pathways? Additional treatment after e.g. ESD for M3 and above, although an R0 ESD resection was achieved? Which therapeutic strategy was used in patients with R0 ESDs but L+ status? Was an additional minimally invasive additional lymphadenctomy considered within the studies reviewed (for instance, M3, R0, L1 patients)? Within the studies reviewed, were any details given regarding the staging prior ESD? Did patients received a CT prior ESD routinely?
Please consider to give further details on study selection criteria.
Author Response
The authors are summarizing up-to-date reports with 5-year survival after endoscopic submucosal dissection (ESD) for superficial esophageal squamous cell carcinoma. Based on the reviewed data the authors conclude, that "The 5-year overall survival rates in m3 or submucosal invasion were relatively low. This discrepancy may be due to the invasiveness of the additional treatment. Additional treatment could be too invasive for the elderly or patients with severe comorbidities. Considering the balance between the risk of metastasis and life expectancy for the elderly or patients with severe comorbidities, the abandonment of additional treatment might be a suitable choice."
This conclusion is not fully supported by the data presented. Details on the criteria for additional treatment of the studies reviewed are partially not given or not discussed. Which criteria were used within the studies for additional treatment? Patient specific individual treatment strategies? standardized pathways? Additional treatment after e.g. ESD for M3 and above, although an R0 ESD resection was achieved? Which therapeutic strategy was used in patients with R0 ESDs but L+ status? Was an additional minimally invasive additional lymphadenectomy considered within the studies reviewed (for instance, M3, R0, L1 patients)? Within the studies reviewed, were any details given regarding the staging prior ESD? Did patients received a CT prior ESD routinely?
Please consider to give further details on study selection criteria.
According to your comment, the conclusion was modified to “additional treatment might be too invasive for the elderly or patients with severe comorbidities. The risk of additional therapy should be balanced against the risk of lymph node metastasis, considering the life expectancy for such patients.”
The Japan esophageal society guideline recommends additional treatment for sm1 or more, or vascular invasion. On the other hand, the European Society of Gastrointestinal Endoscopy (ESGE) guideline recommends additional treatment for sm2 or more, vascular invasion, poorly differentiated tumor, or positive vertical margins. Patients with R0, m3, L1 (vascular invasion) are recommended additional treatment by both Japanese and European guidelines. Patients with R0, sm1, L0 (without vascular invasion) are recommended additional treatment by Japanese guideline, but they are recommended follow-up by European guideline. Roughly speaking, 7 studies were based on Japanese guideline, and 6 studies were based on European guideline among the included 13 studies. According your comment, we added the difference between Japanese and European guidelines and criteria for additional treatment for each study into the revised manuscript.
Minimally invasive additional lymphadenectomy was not performed in the included 13 studies.
Pre-operative examinations were different among the included 13 studies. Both pre-operative endoscopic ultrasound (EUS) and computed tomography (CT) were routinely performed in 5 studies. Pre-operative EUS was routinely performed in 1 study. CT were routinely performed in 1 study. In 2 studies, pre-operative EUS was optional. In 2 studies, pre-operative examination included iodine staining or narrow band imaging (NBI). In 2 studies, pre-operative examination was not described. The information about pre-operative EUS and/or CT was added into the revised manuscript.
The literatures were systematically searched using the PubMed, Cochrane library, and the Igaku-Chuo-Zasshi database in Japan (up to August 2020). The search words were listed below: (esophageal) AND (endoscopic submucosal dissection) AND (squamous cell carcinoma) AND (long-term outcomes). There was no limitation in language. The inclusion criteria were listed below: (1) study design: cohort study; (2) participants: patients who had esophageal squamous cell carcinoma; (3) intervention: endoscopic submucosal dissection (ESD); (4) outcome: 5-year survival. The exclusion criteria were listed below: (1) studies limited to large lesions; (2) 5-year survival including endoscopic mucosal resection (EMR); (3) rescue ESD after chemoradiotherapy (4) meeting abstracts; (5) duplication; and (6) review articles.
As the search results, one hundred fifty-two citations were involved by the literature search process. Among them, we excluded 116 studies according to the exclusion criteria (36 unrelated topics, 63 meeting abstracts, 1 duplication, and 16 review articles). The remaining 36 studies were scrutinized, and 23 studies were rejected (14 studies without 5-year survival, 2 studies limited to large lesions, 5 studies including EMR, 2 rescue ESD after chemoradiotherapy). Due to the limited number of studies comparing ESD and chemoradiotherapy, it was allowed to include EMR for that study. Finally, we included 13 studies in our review. The flowchart showing the searching process is shown in Figure 2. In the revised manuscript, two articles were added (Iizuka et al. in Table 3, and Lee e al. in Table 4).
Thank you for your important comments, which were extremely helpful for improving the quality of our manuscript.